# The Crosstalk of Long Non-Coding RNA and MicroRNA in Castration-Resistant and Neuroendocrine Prostate Cancer: Their Interaction and Clinical Importance

**DOI:** 10.3390/ijms23010392

**Published:** 2021-12-30

**Authors:** Che-Yuan Hu, Kuan-Yu Wu, Tsung-Yen Lin, Chien-Chin Chen

**Affiliations:** 1Institute of Clinical Medicine, College of Medicine, National Cheng Kung University, Tainan 704, Taiwan; greatoldhu@gmail.com; 2Department of Urology, National Cheng Kung University Hospital, College of Medicine, National Cheng Kung University, Tainan 704, Taiwan; hn85386039@gmail.com; 3Division of Urology, Department of Surgery, Dou-Liou Branch, National Cheng Kung University Hospital, College of Medicine, National Cheng Kung University, Yunlin 640, Taiwan; 4Department of Pathology, Ditmanson Medical Foundation Chia-Yi Christian Hospital, Chiayi 600, Taiwan; 5Department of Cosmetic Science, Chia Nan University of Pharmacy and Science, Tainan 717, Taiwan

**Keywords:** cancer, epigenetics, long non-coding RNAs, microRNAs, prostate cancer, castration-resistant, neuroendocrine

## Abstract

Prostate cancer is featured by its heterogeneous nature, which indicates a different prognosis. Castration-resistant prostate cancer (CRPC) is a hallmark of the treatment-refractory stage, and the median survival of patients is only within two years. Neuroendocrine prostate cancer (NEPC) is an aggressive variant that arises from de novo presentation of small cell carcinoma or treatment-related transformation with a median survival of 1–2 years from the time of diagnosis. The epigenetic regulators, such as long non-coding RNAs (lncRNAs) and microRNAs (miRNAs), have been proven involved in multiple pathologic mechanisms of CRPC and NEPC. LncRNAs can act as competing endogenous RNAs to sponge miRNAs that would inhibit the expression of their targets. After that, miRNAs interact with the 3’ untranslated region (UTR) of target mRNAs to repress the step of translation. These interactions may modulate gene expression and influence cancer development and progression. Otherwise, epigenetic regulators and genetic mutation also promote neuroendocrine differentiation and cancer stem-like cell formation. This step may induce neuroendocrine prostate cancer development. This review aims to provide an integrated, synthesized overview under current evidence to elucidate the crosstalk of lncRNAs with miRNAs and their influence on castration resistance or neuroendocrine differentiation of prostate cancer. Notably, we also discuss the mechanisms of lncRNA–miRNA interaction in androgen receptor-independent prostate cancer, such as growth factors, oncogenic signaling pathways, cell cycle dysregulation, and cytokines or other transmembrane proteins. Conclusively, we underscore the potential of these communications as potential therapeutic targets in the future.

## 1. Introduction

Prostate cancer (PCa) is the second most commonly diagnosed malignancy, leading to 6.8% of cancer-related deaths in the male population worldwide in 2020 [1]. Its prevalence increases with age by a 1.7 odds ratio per decade and is up to 59% (48–71%) by age >79 years old [2]. At diagnosis, the treatment and the prognosis of PCa are determined by the stratified risk group on the basis of prostate-specific antigen (PSA) level, the sum of Gleason patterns, and the clinical stage. Compared with localized PCa that can be actively surveyed without management, the treatment of advanced PCa, requiring systemic therapy, is much more complicated and lethal [3,4,5]. Androgen deprivation therapy (ADT) is usually the first-line systemic treatment in advanced PCa, according to the canonical mechanism of the androgen receptor (AR) signaling pathway [6]. Nevertheless, the innate diversity of PCa cells can ultimately adapt to androgen deprivation, resulting in activation of AR signal even under a low level of serum androgen and advancing to a worse condition named castration-resistant prostate cancer (CRPC) [7,8]. 

## 2. CRPC and Its Subtypes

The diagnostic criteria of CRPC consist of serum testosterone less than 50 ng/mL or 1.7 nmol/L plus biochemical or radiologic progression [9]. Biochemical progression is defined as three continuous increases in PSA at intervals of no less than one week, leading to at least two 50% rises over the nadir and PSA more than 2 ng/mL [10]. Radiologic progression is defined as more than two new lesions detected under an imaging survey based on the rules of Response Evaluation Criteria in Solid Tumors (RECIST) [11]. The reported prevalence of CRPC within 5 years of follow-up after initial PCa treatment was 10–20%, with metastatic CRPC (mCRPC) accounting for ≥84% of all. Moreover, 33% non-metastatic CRPC (nmCRPC) advanced to metastasis within 2 years of follow-up [12]. An extensive database series in Sweden in 2020 comprised 1712 CRPC cases, stating that the median PCa-specific survival rates since diagnosis were 30.3 months and 13.3 months for nmCRPC and mCRPC, respectively [13]. 

The mechanism of CRPC is quite complex and alterable, and therefore even if novel therapies are constantly being proposed and used, the median PCa-specific survival in mCRPC can only increase by 2.8 to 4.8 months [14,15,16,17,18,19,20]. Sustained activation of AR signaling in cancer cells even under androgen depletion therapy is thought to be the primary mechanism of CRPC [21]. However, most CRPC are still AR-dependent. AR signaling can still be delivered through several mechanisms, including AR point mutation, increased AR expression, emersion of AR splice variants, altered intratumoral androgen biosynthesis, and co-factor activity modulation [22]. Therefore, more effective AR-targeting agents (AR signaling inhibitor, ARSI) such as abiraterone, apalutamide, and enzalutamide are recognized as first-line treatment options in CRPC [23]. However, not all castration resistance is dependent on AR signaling due to the diversity, plasticity, and heterogeneity of the tumor. In addition, the transition of tumor cells from AR-dependent to AR-independent signaling pathways may cause further resistance to ARSI [24,25]. The mechanisms of AR-independence include neuroendocrine differentiation, cell cycle dysregulation, glucocorticoid receptor upregulation, immune-mediated resistance, PI3K/AKT pathway, and MYC pathway. 

Increasingly more researchers are trying to classify CRPC on the basis of histological phenotypes or by exploring the variation of AR expression, also known as AR heterogeneity in tumor cells. It helps in dissecting the development of CRPC, improving drug evolution and treatment strategy, and delivering more explicit outcomes [26,27,28,29]. Thus far, AR and neuroendocrine markers are the two most common and crucial histologic features helping to categorize CRPC into different phenotypes such as AR-high prostate cancer (ARPC); AR-low prostate cancer (ARLPC); neuroendocrine prostate cancer (NEPC); and both AR-null and NE-null CRPC, also named double-negative CRPC (DNPC) [28,30]. The study published by Bluemn et al. exhibited that through the use of ARSI in the recent two decades, the ratio of NEPC and DNPC in CRPC increased from 5.4% and 6.3% to 13.3% and 23.3%, respectively [29]. The transition from ARPC-dominant phenotype to NEPC-dominant phenotype after receiving AR-targeting therapy has also been demonstrated in animal models, cell lines, and patient-derived xenograft (PDX) studies [31,32,33,34]. In addition to the heterogeneity of cell composition of CRPC, dynamic cell selection and lineage plasticity also increase the complexity of CRPC mechanism and imply the pivotal role of microenvironment, epigenetic or transcriptional, posttranscriptional, and posttranslational regulation.

## 3. Non-Coding RNAs and CRPC

Non-coding RNAs (ncRNAs), which participate in diverse molecular regulatory activities including epigenetic, posttranscriptional, posttranslational regulation, and signal transduction, have been proven to be a crucial molecular substance involved in multiple pathologic mechanisms of CRPC [35,36,37]. In particular, ncRNAs are the transcripted products of genomic sequences that do not proceed to encode proteins. Advanced research and updated knowledge have altered them from useless substances to functional molecules of cellular processes in the past decades [38,39]. Recently, increasing studies have discovered the contribution of ncRNAs on cancer initiation, progression, and metastasis [40]. NcRANs are thought to be potential diagnostic and prognostic biomarkers of cancer, especially combining the detection of different ncRNAs, which can enhance the sensitivity and specificity in diagnostic and prognostic assessments [41]. According to the size, structure, transcripted source, and location, functional ncRNAs can be classified into several subsets, including microRNA (miRNA), small interfering RNA (siRNA or RNAi), small nuclear RNA (snoRNA), PIWI-interactive RNA (piRNA), and the long non-coding RNA (lncRNA). 

Among the different categories of ncRNAs, miRNA and lncRNA are the two most frequently investigated in CRPC research and have been proven to involve multiple pathophysiologic pathways [35,37]. The regulatory role of miRNA in CPRC was the first to be discovered. At least 20 kinds of miRNAs have been confirmed to be related to the development of CPRC, and they participated in diverse pathogenesis pathways such as AR-related cell proliferation, cancer cell survival, apoptosis, or epithelial–mesenchymal transition (EMT) [36]. 

On the other hand, lncRNA is a larger and more complex ncRNA, and it has been gradually explored to be related to CPRC in the past decade due to advances in detection technology [35]. Moreover, because lncRNA carries more information and affinity to biomolecules in the human body, a single type of lncRNA can have multi-pathway regulation. Therefore, lncRNA is also a key regulator in PCa progression from androgen-dependent status to CRPC [42]. For example, AR-regulated long noncoding RNA 1 (ARLNC1, a lncRNA) could be upregulated by the AR protein and stabilize the AR transcript via RNA–RNA interaction [43]. Knockdown of ARLNC1 caused the suppression of AR expression and inhibition of AR-dependent PCa growth in vitro and in vivo [43]. Moreover, prostate cancer-associated non-coding RNA1 (PRNCR1, a lncRNA) and prostate cancer gene expression marker 1 (PCGEM1, a lncRNA) were highly expressed in CRPC cells and reported to be involved in AR signaling pathway, while knockdown of either PRNCR1 or PCGEM1 would suppress in vivo tumor growth of CRPC [44,45,46].

An alternative mechanism for lncRNAs to influence the AR is through interactions with miRNAs. Although the contribution of numerous miRNAs and lncRNA in CRPC development has been summarized and reviewed in the literature, these authors only elaborated on the impact of these two ncRNAs separately without sorting out the interaction of lncRNAs and miRNAs [47,48]. There is accumulating evidence proposed that cancer development is highly correlated with the interaction patterns between lncRNAs and miRNAs [49,50]. LncRNAs and miRNAs are proven to interact in many ways, including the “sponge” effect that is most frequently described and utilized [51,52]. In the last five years, an increasing number of studies have also described the interaction and regulation between lncRNAs and miRNAs in different pathologic pathways of CRPC. Therefore, in this literature review, we comprehensively summarize the currently known interaction between lncRNA and miRNA on the development of CRPC. Further, our collation is classified according to the different molecular mechanisms and histology types of CRPC.

## 4. LncRNA and Its Interaction with miRNA

LncRNAs are defined as non-coding transcripts usually longer than 200 nucleotides. They are one of the most abundant classes within ncRNAs. They outnumber protein-coding transcripts in the ratio of about three to one [53]. In the past few decades, researchers mainly studied the protein-coding genes and their transcripts, yet there are still many unraveled fields in whole transcriptome data. LncRNAs are now found to demonstrate variable effects when binding to DNA, RNA, or proteins. According to their locations corresponding to protein-coding genes, lncRNA can be classified into several groups, including (1) intronic lncRNAs: transcribed from introns of protein-coding genes, (2) intergenic lncRNAs: transcribed between two protein-coding genes, (3) antisense lncRNAs: transcribed in the opposite direction of protein-coding genes, and (4) overlapping lncRNAs: transcripts that cover protein-coding genes [54]. 

LncRNAs can fold themselves into structures to interact with other genomes, transcripts, or proteins to regulate chromatin activities, transcription, protein chromatin assembly, splicing, and telomere biology. They are capable of chromatin remodeling, histone methylation, acting as miRNA “sponges” or transcription factor decoys, and affecting the proteome by mediating complex formation [42]. If lncRNAs are localized in the nucleus, they usually vise proteins to regulate target gene expression at the transcriptional levels. However, if lncRNAs are distributed mainly in the cytoplasm, they are likely to enhance target gene expression at the posttranscriptional level by binding with miRNA.

MiRNAs are small single-stranded ncRNAs with about 22 nucleotides. Most of the time, miRNAs interact with the 3’ untranslated region (UTR) of target mRNAs to repress their expression via the promotion of RNA degradation or inhibition of protein translation. Previously, it has been revealed that 3’UTR of oncogene such as CCND1 can be shortened to escape miRNA-mediated repression in cis and lead to its overexpression in cancer [55]. However, a recent study has demonstrated that 3’UTR shortening of competing for endogenous RNA (ceRNA), an RNA transcript that regulates the levels of other RNA transcripts by competing for shared microRNAs, has a trans effect on the repression of tumor suppressor gene via release of sequestered miRNA and results in tumorigenesis [56]. Taken together, 3’UTR shortening might not only induce proto-oncogene expression in cis but also repress tumor suppressor gene expression in trans via miRNA-dependent manners. This concept of ceRNAs may also be applied to the interactions of lncRNAs and miRNAs and has raised much attention because they can promote or inhibit cancers depending on the roles of the genes they modulate.

In addition, miRNAs have been recognized as well-known effectors concerning castration resistance. The mechanisms of castration resistance related to miRNAs mainly include (1) modulation of AR amplification; (2) interfering AR transcriptional activity; (3) signal cascades independent of AR and crosstalk with AR, or regulation of AR coactivator/corepressor; and (4) AR-targeted miRNAs [35]. Since lncRNAs and miRNAs are two major families of the non-protein-coding transcripts, lncRNAs are prone to act as ceRNAs and competitively bind to miRNAs and create the lncRNA–miRNA–mRNA networks that extensively exist in CRPC cell lines or specimens. Table 1 demonstrates the current studies about the interaction of lncRNAs and miRNAs and their related mechanisms to castration resistance. Among them, signal cascades independent of AR and crosstalk with AR are the most frequently seen and dissected pathways [28]. The AR bypass/crosstalk mechanisms mainly consist of growth factors, cytokines, MAPK, PI3K/AKT, MYC pathway, or cell cycle dysregulation (Table 1, Figure 1) [28].

## 5. LncRNA–miRNA Interaction and Their Roles in AR-Independent/Crosstalk Mechanisms

### 5.1. Growth Factors

Growth factors include insulin-like growth factor-1 (IGF-1), fibroblast growth factor (FGF), and epidermal growth factor (EGF). IGF-1 and insulin-like growth factor 1 receptor (IGF1R) is the most extensively studied growth factor pathway involving AR signaling [73]. IGF1R can interact with AR and modulate the translocation of AR from the cytoplasm to the nucleus and thus alter AR activity [74]. On the other hand, IGF1R can promote AR signaling by stimulating AR coactivators such as transcriptional intermediary factor 2 (TIF2) [74]. Moreover, EGF signal transduction can also activate AR via a ligand-independent pathway [73]. In contrast, FGF activates MAPK signaling and bypasses the AR in promoting CRPC tumor growth by inducing the expression of inhibitor of differentiation 1 (ID1) [29]. 

Since IGF1R, EGFR, and FGFR are receptor tyrosine kinases (RTKs), the role of RTKs and their downstream signal cascade activation may take part in shifting phenotypic and molecular landscapes of metastatic CRPC. A substantial body of evidence has documented that lncRNA–miRNA interactions are involved in these signaling pathways due to their distinctive multifaceted features. Liu et al. assessed the expression of lncRNA AFAP1-AS1 in castration-resistant C4-2, PC3, and DU145 cell lines and found that AFAP1-AS1 level was significantly elevated compared with the androgen-sensitive LNCaP cells. AFAP1-AS1 could bind to miR-15b and disturb its tumor suppressor role to repress the IGF1R [59]. Huang et al. detected that lncRNA PTTG3P was remarkably increased in the castration-resistant cell lines and CRPC tissues. PTTG3P could upregulate PTTG1 expression by competing for miR-146a-3p [60]. In vitro experiments have shown that the PTTG-binding factor directly binds PTTG to promote PTTG nuclear translocation and subsequently increase the expression of FGF mRNA [75].

### 5.2. Oncogenic Signaling Pathways

LncRNA SChLAP1 was found highly expressed in PCa and closely related to tumor progression. SChLAP1 could downregulate the expression of miR-198, and miR-198 had a targeted correlation with 3’ UTR of MAPK1. The miR-198 inhibition significantly increased the expression of MAPK1, and its major target genes such as *ELK-1*, *F-actin*, and *PAK1* are related to cancer progression [61]. 

Upregulated lncRNA Linc00963 was detected in C4-2 cells compared to LNCaP cells, indicating its role in transitioning from androgen-dependent PCa to androgen-independent CRPC [76]. Linc00963 directly bound to miR-655 in CRPC cells and inhibited its interaction with TRIM24 mRNA, promoting cell proliferation by enhancing TRIM24 expression [62]. TRIM24 is a well-known oncogene in advanced CRPC [77]. TRIM24 expression could promote CRPC by activating AR through PI3K/AKT pathway at shallow androgen levels. Additionally, TRIM24 was found to be a transcriptional regulator of EGFR and PIK3CA genes, while PIK3CA and EGFR demonstrated synergetic effects on the PI3K/AKT pathway activation in PCa [62].

LncRNA MYU(VPS9D1-AS1) upregulated the c-Myc transcription level by competitively binding with miR-184 [63]. MYC protein forms a heterodimer with the related transcription factor MAX. This complex acts as transcription factors that activate the expression of multiple proliferative genes through binding to enhancer box sequences and recruiting histone acetyltransferases [78]. Interestingly, MYC is activated by various mitogenic signals such as serum stimulation or by Wnt, Shh, and EGF through the MAPK/ERK pathway [79]. This phenomenon implies the close relationship of the cell surface receptor and intracellular signal networks in promoting PCa cell survival in the absence of androgen or being independent of AR activation.

### 5.3. Cell Cycle Dysregulation

The lncRNA SNHG7 level is also over-expressed in PCa tissue and cell lines, being closely correlated with poor prognosis. Knockdown of SNHG7 significantly decreased the level of cell-cycle regulators, including Cyclin D, CDK4, and CDK6. In vivo and in vitro experiments confirmed the cell cycle arrest at G0/G1 phase after silencing of SNHG7 [64]. In addition, MiR-503 was found to have complementary binding sites with 3’UTR of SNHG7 and Cyclin D. Transfection with miR-503 mimics could downregulate the SNHG7 and Cyclin D mRNA expression and suppress tumor growth [64].

LncRNA LOXL1-AS1 was predominantly located in the PCa cell cytoplasm and functioned as a miRNA sponge to interact with miR-541-3p [65]. MiR-541-3p could inhibit Cyclin D expression by binding with its 3’UTR and induce G0/G1 phase cell cycle arrest. LOXL1-AS1 induced upregulated genes significantly enriched in cell cycle checkpoint and nuclear division. Collectively, LOXL1-AS1 regulated cell cycle progression through modulating the expression of miR-541-3p and, subsequently, the expression of Cyclin D [65]. 

Y chromosome-transcribed lncRNA TTTY15 was also mainly located in the cytoplasm and acted as a ceRNA to absorb the let-7 family members let-7a, let-7b, let-7c, and let-7f. Among the let-7 targets, CDK6 and fibronectin (FN1) were of clinical interest. CDK6 regulates cell proliferation, and FN1 plays a vital role in cancer metastasis [80]. Furthermore, FN1 has been shown to stimulate the gonadal steroids that interact with vertebrate ARs, capable of controlling the expression of Cyclin D and related genes involved in cell cycle control. Taken together, TTTY15 may promote PCa progression by sponging let-7 family members and enhancing CDK6 and FN1 expression [66]. As a monoallele on the Y chromosome, the *TTTY15* gene has a unique advantage for gene editing therapy and thus may possess more therapeutic potential. Lastly, Figure 2 summarizes current molecular shreds of evidence of lncRNAs and miRNAs in the cell cycle progression of CRPC.

### 5.4. Cytokines and Other Transmembrane Proteins

Integrins are transmembrane proteins that facilitate cell-to-cell and cell-to-extracellular matrix adhesion [81]. CD51, otherwise named integrin subunit alpha V (ITGAV), is a member of the integrin alpha chain family [82]. CD51 inhibited E-cadherin expression and activated transforming growth factor-beta (TGF-β), a well-known multifunctional cytokine [83]. Mechanical tension of the cells alongside binding of the large latent TGF-β complex exposes the active TGF-β through the latency-associated peptide (LAP) opening. By binding to TGF-β receptor on neighboring cells, activated TGF-β acts on the surrounding stromal, immune, endothelial, and smooth muscle cells. It results in immunosuppression and angiogenesis, finally enhancing the invasiveness of cancer cells [84]. LncRNA SNHG17 level is elevated in CRPC cell lines and tissues and is positively correlated with CD51 expression. Mechanically, SNHG17 directly bound with miR-144 and inhibited miR-144 targeting downstream CD51. Herein, SNHG17 enhanced cell proliferation and invasion in vitro and promoted tumor growth and metastasis in vivo [67]. 

## 6. The Interplay between lncRNA and miRNA/Epigenetic Regulators in NEPC

### 6.1. General Introduction of NEPC

NEPC is a rare subtype of prostate cancer. From 2004 to 2013, only 0.06% of patients were classified to de novo NEPC among patients diagnosed with prostate cancer in the Surveillance, Epidemiology, and End Results (SEER) database [85]. However, with the advancement of AR-targeting therapy, NEPC can be detected more and more nowadays. Compared to de novo NEPC, patients who received ARSI may develop NEPC, called treatment-emergent neuroendocrine prostate cancer (t-NEPC). The t-NEPC is aggressive and lethal. Clinically, t-NEPC was found to have a higher visceral organs metastatic rate, a higher tumor cell proliferation rate, a higher resistance rate of ADT, and a poor prognosis [86]. Moreover, t-NEPC is AR-independent with negativity for PSA, and its prevalence rate is about 10~20% among patients who developed mCRPC [87]. Therefore, the way in which to treat t-NEPC has become an important issue in the era of ARSI. The mechanism of NEPC development is still limited knowledge today. There are two possible proposals as follows: (1) genomic mutation, (2) epigenetic/non-coding RNA interactome (ENI). We describe them separately here.

### 6.2. Genomic Mutation and NEPC

In recent evidence, retinoblastoma (Rb), p53, PTEN, AURKA, MYCN, and SRRM4 were found to be associated with NEPC. Loss of Rb gene, p53, and PTEN was present in more than 50% of NEPC [32,88,89,90]. Inactivation of Rb and p53 in PCa may increase the expression of SOX2, which is a reprogramming transcription factor, thus inducing stem cell-like status and promoting lineage plasticity [33]. Moreover, amplification and overexpression of AURKA and MYCN were present above 70% in NEPC but only 5% in ordinary PCa [91]. Moreover, N-Myc and AKT1 induce adenocarcinoma from prostate epithelial cells into neuroendocrine tumors [92]. Notably, overexpression of SRRM4 was also found in NEPC and might induce NEPC development by SOX2 expression, which promotes stem cell-like status and lineage plasticity [93,94]. Moreover, SRRM4 targets repressor element-1 silencing transcription factor (REST), which drives neuroendocrine transdifferentiation [95]. In brief, loss of tumor suppressor genes (such as *Rb*, *p53*, and *PTEN*) or protooncogene activation (such as AURKA, MYCN, and SRRM4) may induce lineage plasticity and drive NEPC development.

### 6.3. ENI and NEPC

Emerging evidence has revealed t-NEPC development from PCa by transdifferentiation of luminal cells into neuroendocrine cells. It was described in many studies that epigenetic alterations might drive NEPC formation. Moreover, EZH2, SOX2, REST, FOXA1, FOXA2, BRN2, ONECUT2, E2F1, and ASCL1 were found to be associated with NEPC progression. In NEPC, Rb inactive or N-Myc activation can induce EZH2 overexpression, promoting t-NEPC development [96,97]. In addition, ADT can induce CREB (cAMP response element-binding protein)-EZH2-TSP1 (thrombospondin-1) pathway, which promotes neuroendocrine differentiation and angiogenesis in advanced PCa [98]. Moreover, N-Myc interacts with EZH2 and promotes the reprogramming of cells into a stem-like cell status. This step may speed up neuroendocrine differentiation in advanced PCa [97]. As mentioned above, SOX2 overexpression promotes cancer stem-like cell formation and lineage plasticity and NEPC progression, regulated by Rb, p53, and SRRM4 [33,94]. Moreover, REST, also known as neuron-restrictive silencer factor (NRSF), is a transcriptional repressor of neuronal differentiation. REST influences neuroendocrine transdifferentiation, associated with SRRM4 expression and epithelial–mesenchymal transition (EMT) [95,99]. The association between EMT and cancer stem-like cells was demonstrated in previous reports [100]. The crosstalk between the WNT–β-catenin and IL-6-STAT3 pathways was found in EMT/cancer stem-like cells in cancer cells [101]. It is reported that activation of the EMT may induce cellular-plasticity, stem-like cell status, and neuroendocrine differentiation in advanced Pca [102]. The FOXA family, including FOXA1 and FOXA2, is also associated with neuroendocrine differentiation in PCa. The high expression of FOXA2 was found in NEPC [103,104]. By contrast, FOXA1 inhibits neuroendocrine differentiation via IL-8 and the MAPK/ERK pathway in PCa [105]. In short conclusion, the overexpression of EZH2, SOX2, FOXA2, BRN2, ONECUT2, E2F1, and ASCL1 but the downregulation of FOXA1 and REST may promote NEPC development [106,107,108,109,110].

### 6.4. The Interplay between lncRNA and miRNA in NEPC

The interplay between lncRNA and miRNA/epigenetic regulators in NEPC is described in Table 2. According to previous studies, lncRNA including HOTAIR, H19, LINC00261, FENDRR, and MALAT1 could crosstalk with miRNA in NEPC. For example, HOTAIR and miR-31-5, LINC00261 and miR-8485, H19 and miR-675, FENDRR and miR-301b-3p/miR-18a-5p, and MALAT1 and miR-1 have been reported [111,112,113,114,115,116,117].

Notably, it was proposed that there was a direct interaction network among AR, HOTAIR, ESR1, and miR-31-5p. It was found in NEPC that miR-31-5p inhibits the expression of AR, while AR transcriptionally inhibits HOTAIR [111]. Otherwise, HOTAIR is a REST-repressed lncRNA that drives NEPC development [112]. Interestingly, AR itself, in turn, can repress miR-31 transcription, and HOTAIR can enhance both AR and ESR1 [111]. The four molecules form a mutual regulatory model in NEPC.

Mather et al. reported that a highly conserved lncRNA, LINC00261, would bind with miR-8485 to reduce its inhibition to chromobox 2 (CBX2) in the cytoplasm [113]. *CBX2* overexpression can promote tumor proliferation, while the overexpression of miR-8485 can inhibit the expression of *CBX2*. On the other hand, LINC00261 actives forkhead box A2 (FOXA2) expression via SMAD2/3 transcriptional complex in the nucleus, while FOXA2 overexpression may induce neuroendocrine differentiation. 

Zhu et al. demonstrated lncRNA H19 suppresses tumor cell migration via miR-675. MiR-675 directly binds to the 3’UTR of TGFβ1 mRNA that inhibits its translocation to inhibit tumor migration [114]. In addition, FENDRR, also known as FOXF1-AS1, can reduce tumor invasion by targeting CSNK1E in PCa. Meanwhile, miR-301b-3p plays a role in upregulatory tumor progression, although the interaction is not clear between FENDRR and miR-301b-3p [115]. Moreover, it was proposed that FENDRR inhibits tumor cell proliferation and migration by binding to miR-18a-5p with RUNX1 [116].

Recently, the association between MALAT1 and NEPC was demonstrated by Ostano et al. [111]. Although the mechanism between MALAT1 and miRNA in NEPC remains unclear, the relation between MALAT1 and miR-1 in AR-independent PCa cell lines was shown. In PCa cell lines, MALAT1 expression was upregulated, while miR-1 expression was downregulated. Chang et al. proposed the idea that MALAT1 acts as a molecular sponge of miR-1, resulting in downregulating KRAS in AR-independent PCa cell lines [117]. In this way, tumor cell proliferation and migration are suppressed by silencing MALA1. 

In addition to the above, there were other lncRNAs interacting with epigenetic regulators in NEPC, including lncRNA-p21, MIAT, LINC00514, and SSTR5-AS1. For example, lncRNA-p21 interacts with EZH2 and disrupts the PRC2 complex. Then, lncRNA-p21 promotes EZH2 to enhance STAT3 methylation and drive neuroendocrine transdifferentiation. This phenomenon was observed when patients were treated with enzalutamide [118]. Crea et al. demonstrated that MIAT interacts with polycomb genes and is positively associated with higher metastatic potential, *Rb* mutation, and *CBX2* expression [119]. LINC00514 links with TADA3 and inhibits its activity in NEPC, while TADA3 can bind and stabilize p53 [120,121]. Regarding SSTR5-AS1, it is the highest expressed lncRNA in NEPC and binds with KDM4B, a histone demethylase and a key molecule in AR signaling [120]. In addition, KDM4B influences AR transcriptional activity in prostate adenocarcinoma and interacts with N-Myc in neuroblastoma [122,123]. Because N-Myc plays a vital role in neuroendocrine transdifferentiation, KDM4B may drive NEPC via N-Myc [120].

## 7. Conclusions

In recent years, ncRNA biology has been regarded as an important role in our understanding of PCa progression, especially in the era of AR-targeting agents. In order to deal with a complete diminishment of direct AR activation, cancer cells have started learning to “survive” in the environment poverty of AR signaling. As a member of the steroid hormone nuclear receptor family, AR can act as a nuclear transcription factor with its coactivators and directly binds to androgen response elements (AREs) to mediate its target gene function. However, cancer cells can still bypass AR-binding-independent pathways via activation of second messengers such as MAPK, ERK, or AKT. Growth factors or cytokines are also crucial for PCa to develop castration resistance.

On the other hand, due to the widespread use of AR-targeting agents, the prevalence of NEPC also increased in decades. LncRNA–miRNA–mRNA networks appear to be widespread in PCa cells and tissues. LncRNAs are also recognized as master epigenetic regulators of the genome in the development of NEPC. Furthermore, an investigation into the mechanisms of ncRNA actions is a rising issue due to the relative accessibility to construct their mimics and put them into therapeutic potential. Methods are now available for interfering lncRNAs, but the application of these approaches is still immature. However, our current literature review rendered a curation primarily focused on the interaction of lncRNA and miRNA in CRPC and NEPC. More attention and research in this field are warranted.

## Figures and Tables

**Figure 1 ijms-23-00392-f001:**
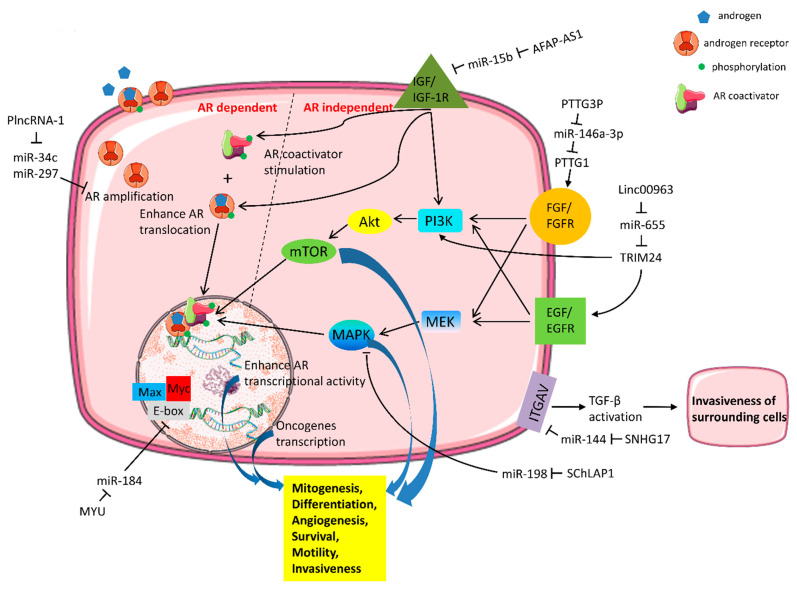
The mechanisms of actions for long-non-coding RNA (lncRNA) and microRNA (miRNA) in androgen receptor (AR) independence/crosstalk signaling in castration-resistant prostate cancer cells. A number of miRNAs that are regulated by lncRNAs and involve the activation of AR crosstalk (AR dependent) and AR bypass (AR independent) signaling axis are demonstrated and further discussed in the present manuscript.

**Figure 2 ijms-23-00392-f002:**
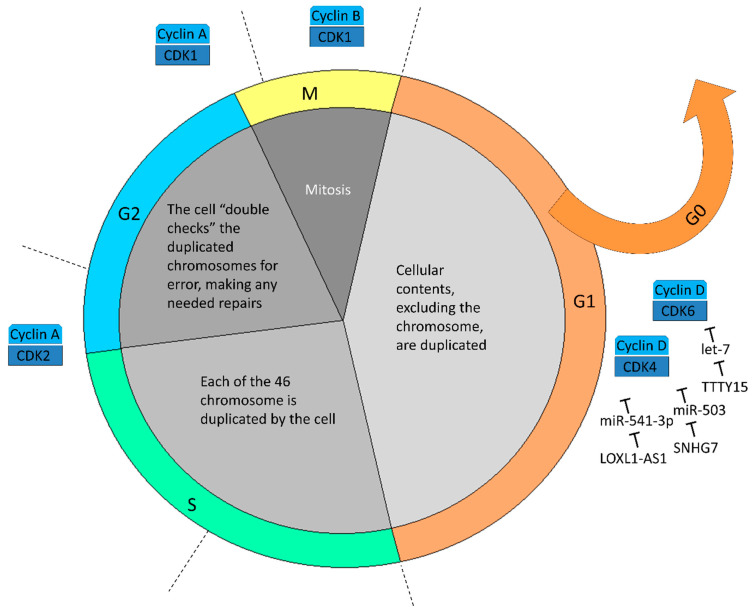
The mechanisms of actions for lncRNAs and miRNAs in cell cycle progression in castration-resistant prostate cancer cells. Most of the miRNAs in contemporary pieces of evidence function in the canonical checkpoint that prevents G1 phase cells from entering the S phase. Cancer cells overwhelm pRb growth suppression through inactivation of pRb by Cyclin D-CDK4/6. MiRNAs inhibit Cyclin D-CDK4/6, but their inhibitory ability is abolished by lncRNAs indicated above.

**Table 1 ijms-23-00392-t001:** Content curation of long-non-coding RNA (lncRNA) and its targeting microRNA (miRNA) and related mechanism of castration resistance prostate cancer.

Mechanism of Castration Resistance	Subject Investigated (Cell Lines/Tissue)		lncRNA	miRNA	Interactions	Reference
AR amplification	22RV1/human PCa		PlncRNA-1	miR-34c/miR-297	PlncRNA-1 promotes AR expression via competitive inhibition of miRNA-34c/miR-297 targeting AR.	[57]
Increased AR transcriptional activity	PC3, DU145/human PCa		CCAT1	miR-28-5P	In cytoplasm: competing for miR-28-5P to promote cell proliferation and colony formation.	[58]
					In nucleus: CCAT1 acts as a scaffold for DDX5(P68) and AR transcriptional complex to facilitate expression of AR-regulated castration resistance gene (*UBE2C*).	
Signal cascades independent of AR and crosstalk with AR	Growth factors	C4-2, PC3, DU145	AFAP1-AS1	miR-15b	AFAP1-AS1 upregulates IGF1R by competitively binding with miR-15b to de-repress IGF1R.	[59]
		human PCa	PTTG3P	miR-146a-3p	PTTG3P upregulates PTTG1 to stimulate FGF expression by competing for miR-146a-3p.	[60]
	Other oncogenic signal pathways	PC3	SChLAP1	miR-198	SChLAP1 regulates the miR-198 expression and influences cancer progression by the MAPK1 pathway.	[61]
		human PCa	Linc00963	miR-655	Linc00963 competitively binds with miR-655 and upregulates TRIM24 expression to activate the PI3K/AKT pathway.	[62]
		PC3, DU145/human PCa	MYU(VPS9D1-AS1)	miR-184	MYU upregulates the MYC expression by competitively binding with miR-184.	[63]
	Cell cycle dysregulation	DU145/human PCa	SNHG7	miR-503	MiR-503 targets 3′-UTR of SNHG and inhibits cell cycle proteins (CDK4, CDK6, Cyclin D), inducing G0/G1 cell cycle arrest.	[64]
		PC3, DU145	LOXL1-AS1	miR-541-3p	LOXL1-AS1 interferes with miR-541-3p targeting cell cycle regulator Cyclin D and promotes cell proliferation.	[65]
		DU145/human PCa	TTTY15	let-7	FOXA1, acting as a transcription factor of TTTY15, promotes PCa progression by sponging let-7 and upregulating CDK6 and FN1.	[66]
	Cytokine	C4-2, PC3, DU145	SNHG17	miR-144	SNHG17 acts as a ceRNA to upregulate CD51 (integrin alpha-V) expression through competitively sponging miR-144.	[67]
Unidentified mechanisms	PC3, DU145		HOTAIR	miR-34a	MiR-34a directly targets HOTAIR and inhibits cell growth.	[68]
	PC3		HOTAIR	miR-193a	HOTAIR couples with EZH2 to repress miR-193a by trimethylation of H3K27me3; miR-193a directly targets HOTAIR to reduce HOTAIR level in miR-193a overexpressed cells.	[69]
	PC3		PCGEM-1	miR-148a	Putative PCGEM1 binding site is identified in the 5′-UTR of miR-148a; PCGEM-1 expression represses miR-148a and cell apoptosis.	[70]
	PC3, DU145/human PCa		PCSEAT	miR-143-3p-/miR-24-2-5p	PCSEAT competitively sponges miR-143-3p/miR-24-2-5p and decreases PCSEAT-mediated cell proliferation.	[71]
	PC3, DU145/human PCa		Linc00308	miR-137	LncRNA-miRNA-mRNA networks regulate tumor suppressor gene *PTGS2* and *DUSP2* and affect survival.	[72]
			Linc00355	miR-122/miR-506		
			OSTN-AS1	miR-137/miR-506		

AR: Androgen receptor; FGF: fibroblast growth factor; FN1: fibronectin; IGF1R: insulin-like growth factor 1 receptor; PCa: prostate cancer; ceRNA: competing endogenous RNA.

**Table 2 ijms-23-00392-t002:** The interplay between lncRNA and miRNA/epigenetic regulators in NEPC.

The Interplay between lncRNA and miRNA				
lncRNA	miRNA	Target	Pathway and Influence	Reference
HOTAIR	miR-31-5p	REST, EZH2	A direct interaction network among AR, HOTAIR, ESR1, and miR-31-5p was proposed. MiR-31-5p inhibits the expression of AR, and then AR transcriptionally inhibits HOTAIR.	[111,112]
LINC00261	miR-8485	CBX2, FOXA2	In the cytoplasm, LINC00261 binds to miR-8485, which reduces the inhibition of CBX2. In the nucleus, LINC00261 activates FOXA2 expression via the SMAD2/3 transcriptional complex.	[113]
H19	miR-675	TGF-β1	H19 and miR-675 negatively regulate the expression of TGFβ1, inhibiting prostate cancer migration.	[114]
FENDRR (FOXF1-AS1)	miR-301b-3p	CSNK1E, PRC2	Reduce tumor invasion by targeting CSNK1E.	[115]
	miR-18a-5p	RUNX1	FENDRR inhibits tumor cell proliferation by binding to miR-18a-5p with RUNX1.	[116]
MALAT1	miR-1		MALAT1 acts as a sponge of miR-1, resulting in downregulating KRAS in AR independent prostate cancer.	[117]
**The interplay between lncRNA and epigenetic regulators**				
LncRNA-p21		EZH2	LncRNA-p21 promotes EZH2 to enhance STAT3 methylation and drives neuroendocrine transdifferentiation.	[118]
MIAT		Polycomb genes	MIAT interacts with polycomb genes and is positively associated with *Rb* mutation and *CBX2* expression.	[111,119]
LINC00514		TADA3	LINC00514 interacts with TADA3 and reduces p53 activity.	[120]
SSTR5-AS1		KDM4B	SSTR5-AS1 binds with KDM4B. KDM4B interacts with N-Myc, which drives the progression of NEPC.	[120]

## Data Availability

The data that support the findings of this study are available from the corresponding author upon reasonable request.

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
