# Peer review of "The Crosstalk of Long Non-Coding RNA and MicroRNA in Castration-Resistant and Neuroendocrine Prostate Cancer: Their Interaction and Clinical Importance"

_ijms, 2021, doi:10.3390/ijms23010392_

Round 1

Reviewer 1 Report

Title of review article:

 „The Crosstalk of Long Non-coding RNA and MicroRNA in Castration-resistant and Neuroendocrine Prostate Cancer: Their Interaction and Clinical Importance.“ is appropriate and may fit the Journal topic.

 The paper contains a number of significant omissions, which do not qualify it for publication: There is widespread literature agreement that „The mechanism of CRPC is quite complex and alterable, so even if novel therapies are constantly being proposed and used. Most CRPC are still AR-dependent. As concerns the epigenetic regulation of prostate carcinoma development, the text is extensive, but nothing new has been documented. It gives only a recapitulation of previous results, views and hypotheses, only the elementary things are repeated concerning: Non-coding RNAs and CRPC and LncRNA and its interaction with miRNA.The any contribution is a systematic presentation of table 1: „Content curation of long-non-coding RNA (lncRNA) and its targeting microRNA (miRNA) and related mecha-nism of castration resistance prostate cancer“. The part 5 „LncRNA-miRNA interaction and their roles in AR independent/crosstalk mecha-nisms“ starts with general rrole of growth factors in cancerogenesis“, oncogenes, cell cycle dysregulation, cytokines signaling in cancerogenesis... completely moves away from the main focus, the LncRNA and miRNA interactions in CRPC.There is no any novel knowledge about the role of Cancer stem cells (CSC) and their role in the self-renewal and differentiation.There is no any hypothesis about the shortening of the UTR repression sites of mRNA as a main transcriptional remodeling of PCA.There are no any aspects of the theranostic potential of ncRNAs what may be of clinical importance what the title may anticipate or propose.

After the careful analysis of the papers cited in proposed manuscript, it can be concluded that there are no papers on this topic written by the proposed authors (or perhaps only one was once a co-author? This may suggest that the authors are still in process of studying the mechanism of action and the role of ncRNAs. Carreful literature review represents only an initial step and it is far from being able to suggest future guidelines in theranostics.

Critically appraising other already published works related to non-codding RNAs in prostate cancer, all of above mentioned remarks makes the manuscript of any novelity. Maintaining a high standard of current journal, the paper does not meet that standard and it should be rejected.

Rejected

Title of review article:

 „The Crosstalk of Long Non-coding RNA and MicroRNA in Castration-resistant and Neuroendocrine Prostate Cancer: Their Interaction and Clinical Importance.“ is appropriate and may fit the Journal topic.

 The paper contains a number of significant omissions, which do not qualify it for publication: There is widespread literature agreement that „The mechanism of CRPC is quite complex and alterable, so even if novel therapies are constantly being proposed and used. Most CRPC are still AR-dependent. As concerns the epigenetic regulation of prostate carcinoma development, the text is extensive, but nothing new has been documented. It gives only a recapitulation of previous results, views and hypotheses, only the elementary things are repeated concerning: Non-coding RNAs and CRPC and LncRNA and its interaction with miRNA.The any contribution is a systematic presentation of table 1: „Content curation of long-non-coding RNA (lncRNA) and its targeting microRNA (miRNA) and related mecha-nism of castration resistance prostate cancer“. The part 5 „LncRNA-miRNA interaction and their roles in AR independent/crosstalk mecha-nisms“ starts with general rrole of growth factors in cancerogenesis“, oncogenes, cell cycle dysregulation, cytokines signaling in cancerogenesis... completely moves away from the main focus, the LncRNA and miRNA interactions in CRPC.There is no any novel knowledge about the role of Cancer stem cells (CSC) and their role in the self-renewal and differentiation.There is no any hypothesis about the shortening of the UTR repression sites of mRNA as a main transcriptional remodeling of PCA.There are no any aspects of the theranostic potential of ncRNAs what may be of clinical importance what the title may anticipate or propose.

After the careful analysis of the papers cited in proposed manuscript, it can be concluded that there are no papers on this topic written by the proposed authors (or perhaps only one was once a co-author? This may suggest that the authors are still in process of studying the mechanism of action and the role of ncRNAs. Carreful literature review represents only an initial step and it is far from being able to suggest future guidelines in theranostics.

Critically appraising other already published works related to non-codding RNAs in prostate cancer, all of above mentioned remarks makes the manuscript of any novelity. Maintaining a high standard of current journal, the paper does not meet that standard and it should be rejected.

Rejected

Author Response

The Crosstalk of Long Non-coding RNA and MicroRNA in Castration-resistant and Neuroendocrine Prostate Cancer: Their Interaction and Clinical Importance.“ is appropriate and may fit the Journal topic.

  1. There is widespread literature agreement that „The mechanism of CRPC is quite complex and alterable, so even if novel therapies are constantly being proposed and used. Most CRPC are still AR-dependent. As concerns the epigenetic regulation of prostate carcinoma development, the text is extensive, but nothing new has been documented. It gives only a recapitulation of previous results, views and hypotheses, only the elementary things are repeated concerning: Non-coding RNAs and CRPC and LncRNA and its interaction with miRNA.The any contribution is a systematic presentation of table 1: „Content curation of long-non-coding RNA (lncRNA) and its targeting microRNA (miRNA) and related mecha-nism of castration resistance prostate cancer“. 
  • Thank you for your comments. We agree CRPC is complex and challenging in its treatment. Since this article is an updated “review” not an original article or perspective editorial, we cannot propose or give something new without experimental data. As you said, CRPC is complex and its epigenetic pathogenesis is still unclear, although it’s widely investigated. Herein, we could only systematically organize the updated published articles and comprehensively write the epigenetic connections logically.
  • Although the elementary things (Non-coding RNAs and CRPC and LncRNA and its interaction with miRNA) are repeatedly concerning in our review, it is a pity that the current shreds of evidence still have no significant progress in the epigenetic pathogenesis of CRPC and neuroendocrine prostate cancer (NEPC). Therefore, we humbly believe that reviewing updated articles may help facilitate our understandings of this field. Moreover, we are grateful for the praises in Table 1.
  • Compared to other reviews, our review additionally focuses on the interplay of lncRNAs and miRNA in NEPC, besides CRPC. The readers can compare the epigenetic involvement in two groups of aggressive prostatic cancers.
  1. The part 5 „LncRNA-miRNA interaction and their roles in AR independent/crosstalk mecha-nisms“ starts with general rrole of growth factors in cancerogenesis“, oncogenes, cell cycle dysregulation, cytokines signaling in cancerogenesis... completely moves away from the main focus, the LncRNA and miRNA interactions in CRPC.
  • Thank you for your comments. As the reviewer may know, lncRNAs and miRNAs modulate mRNA expression, thereby controlling multiple cellular processes, including transcriptional regulation of gene expression, cell differentiation, and apoptosis. Since the readers of review articles are the general population, we need first to introduce the growth factor, hereby oncogenic signal pathway, cell cycle dysregulation, and cytokines in part 5. It is a logical way to help prostate surgeons or oncologists figure out the field. We hope the reviewer can understand that review articles written by specialists are set for a better understanding in some fields to public readers and researchers.
  1. There is no any novel knowledge about the role of Cancer stem cells (CSC) and their role in the self-renewal and differentiation.
  • Thank you for your professional recommendation. Since this review focuses on the crosstalk of lncRNAs and miRNAs, we did not write too much about cancer stem cells to avoid running off the topic. To follow the recommendation, we have added some discussions in parts 6.2 and 6.3 on page 9. All revised parts were highlighted in red color in the revised manuscript. We are grateful for your sincere comments.
  1. There is no any hypothesis about the shortening of the UTR repression sites of mRNA as a main transcriptional remodeling of PCA.
  • Thank you for your comments. With updated references, we have added the issue in part 4 on page 4. All revised parts were highlighted in red color in the revised manuscript. We are grateful for your sincere comments.
  1. There are no any aspects of the theranostic potential of ncRNAs what may be of clinical importance what the title may anticipate or propose.
  • Thank you for your professional recommendation. Although there are no clinical trials of therapeutic agents derived from lncRNAs or miRNAs in prostate cancer, we cannot ignore the therapeutic potential. According to the suggestion, we have added the discussion on page 3. All revised parts were highlighted in red color in the revised manuscript. We are grateful for your sincere advice.
  1. After the careful analysis of the papers cited in proposed manuscript, it can be concluded that there are no papers on this topic written by the proposed authors (or perhaps only one was once a co-author? This may suggest that the authors are still in process of studying the mechanism of action and the role of ncRNAs. Carreful literature review represents only an initial step and it is far from being able to suggest future guidelines in theranostics.
  • Thank you for your comments. The authors are prostate surgeons and pathologists with extensive experience in prostatic cancer treatment and diagnosis. We may have limited articles in the epigenetic investigation of prostatic cancer, but our research articles in cancer biology are over 50. Since the studies of lncRNA and miRNA are still developing, and most epigenetic mechanisms are still unclear, we agree that “we are still in the process of studying the mechanism of action and the role of ncRNAs.” It is a lifetime work to investigate the epigenetics and carcinogenesis in prostate cancer. We are humbly invited by IJMS editorial office and write this updated review which may be too basic for top experts like you. However, we hope the reviewer can figure out our efforts. Lastly, the guideline of IJMS suggests extensive and updated literature review for a review article, but “Theranostics” is another journal.

Reviewer 2 Report

Accept as it is

I strongly believe that this paper can be presented in the received form. It is a review that gives authors freedom in the form of writting and the order of stating scientific facts. Basded on my knowledge and expertise, i am of the opinion that the autohrs have written a comprehensibe,up to date, review.  That was the reason for my decision Accept as it is.

Author Response

Reviewer 2:

I strongly believe that this paper can be presented in the received form. It is a review that gives authors freedom in the form of writting and the order of stating scientific facts. Basded on my knowledge and expertise, i am of the opinion that the autohrs have written a comprehensibe,up to date, review.  That was the reason for my decision Accept as it is.

  • Thank you for your kind words and professional recommendation. We’re grateful for your sincere comments.

Round 2

Reviewer 1 Report

The article has been improved, but not in a manner to be published in given form.

1)The abstract was not improved anymore. In this form it does not promise any  novelty.

2) The sentences: "In cancer, oncogenes in cis utilize 3’ UTR shortening to escape miRNA repression. Otherwise, 3’ UTR shortening may also attenuate their role as a miRNA sequester."I am afraid there are not correct statements:  the non-coding RNAs availability can profoundly alter the translation rate, because the extended UTR regions contain high numbers of miRNA-silencing targets, acting as the translational attenuators, suitable for potential pairing of miRNas.  From the other side, the UTR shortening in the mRNAs of proto-oncogenic proteins may led to their transformation into oncogenic proteins, because they can be translated without repressive control of miRNas. Therefore, the shortening of mRNA UTR region (polyadenylation code) occurs by the enzymatic action of different RNases (PARN and others RNases od D type), not by miRNAs, which are regulated by many different mRNA binding proteins.

Author Response

Reviewer 1:

The article has been improved, but not in a manner to be published in given form.

  1. The abstract was not improved anymore. In this form it does not promise any novelty.. 
  • Thank you for your sincere comments. We have re-written the abstract and emphasized our aims and critical points in this review. All revised parts were highlighted in red color in the revised manuscript. We are grateful for your professional recommendation.
  1. The sentences: "In cancer, oncogenes in cis utilize 3’ UTR shortening to escape miRNA repression. Otherwise, 3’ UTR shortening may also attenuate their role as a miRNA sequester." I am afraid there are not correct statements: the non-coding RNAs availability can profoundly alter the translation rate, because the extended UTR regions contain high numbers of miRNA-silencing targets, acting as the translational attenuators, suitable for potential pairing of miRNas. From the other side, the UTR shortening in the mRNAs of proto-oncogenic proteins may led to their transformation into oncogenic proteins, because they can be translated without repressive control of miRNas. Therefore, the shortening of mRNA UTR region (polyadenylation code) occurs by the enzymatic action of different RNases (PARN and others RNases od D type), not by miRNAs, which are regulated by many different mRNA binding proteins.
  • Thank you for your professional comments. According to your recommendation, we have corrected and re-written this discussion on page 4 as follows, “MiRNAs are small single-stranded ncRNAs with about 22 nucleotides. Most of the time, miRNAs interact with the 3’ untranslated region (UTR) of target mRNAs to repress their expression via the promotion of RNA degradation or inhibition of protein translation. Previously, it has been revealed that 3’UTR of oncogene such as CCND1 can be shortening to escape miRNA-mediated repression in cis and lead to its overexpression in cancer [51]. However, a recent study has demonstrated that 3’UTR shortening of competing for endogenous RNA (ceRNA), an RNA transcript that regulates the levels of other RNA transcripts by competing for shared microRNAs, has a trans effect on the repression of tumor suppressor gene via release of sequestered miRNA and results in tumorigenesis [52]. Taken together, 3’UTR shortening might not only induce proto-oncogene expression in cis but also repress tumor suppressor gene expression in trans via miRNA-dependent manners. This concept of ceRNAs may also be applied to the interactions of lncRNAs and miRNAs and has raised much attention because they can promote or inhibit cancers depending on the roles of the genes they modulate.”
  • All revised parts were highlighted in red color in the revised manuscript. We appreciate your sincere comments and expertise.
